



# Three-dimensional radiative transfer effects on airborne, satellite and ground-based trace gas remote sensing

Marc Schwaerzel[1,2], Claudia Emde[3], Dominik Brunner[1], Randulph Morales[1], Thomas Wagner[4], Alexis Berne[2], Brigitte Buchmann[1], and Gerrit Kuhlmann[1]

[1]Empa, Swiss Federal Laboratories for Materials Science and Technology, Dübendorf, Switzerland
[2]Environmental Remote Sensing Laboratory, École Polytechnique Fédérale de Lausanne, Lausanne, Switzerland
[3]Meteorological Institute, Ludwig-Maximillians-University, Munich, Germany
[4]Max-Planck-Institute for Chemistry, Mainz, Germany

**Correspondence:** Gerrit Kuhlmann (gerrit.kuhlmann@empa.ch)

**Abstract.** Air mass factors (AMF) are used in passive trace gas remote sensing for converting slant column densities (SCD) to vertical column densities (VCD). AMFs are traditionally computed with 1D radiative transfer models assuming horizontally homogeneous conditions. However, when observations are made with high spatial resolution in a heterogeneous atmosphere or above a heterogeneous surface, 3D effects may not be negligible. To study the importance of 3D effects on AMFs for different

types of trace gas remote sensing, we implemented 1D-layer and 3D-box AMFs into the Monte Carlo radiative transfer model (RTM) MYSTIC. The 3D-box AMF implementation is fully consistent with 1D-layer AMFs under horizontally homogeneous conditions and agrees very well (<5% relative error) with 1D-layer AMFs computed by other RTMs for a wide range of scenarios. 3D-box AMFs make it possible to visualize the 3D spatial distribution of the sensitivity of a trace gas observation, which we demonstrate with two examples. First, we computed 3D-box AMFs for ground-based multi-axis spectrometer (MAX-

DOAS) observations for different viewing and aerosol scenarios. The results illustrate how the sensitivity reduces with distance from the instrument and that a non-negligible part of the signal originates from outside the line of sight. Such information is invaluable for interpreting MAX-DOAS observations in heterogeneous environments such as urban areas. Second, 3D-box AMFs were used to generate synthetic nitrogen dioxide ($NO_2$) SCDs for an airborne imaging spectrometer observing the $NO_2$ plume emitted from a tall stack. The plume was imaged under different solar zenith angles and solar azimuth angles.

To demonstrate the limitations of classical 1D-layer AMFs, VCDs were then computed assuming horizontal homogeneity. As a result, the imaged $NO_2$ plume was shifted in space, which led to a strong underestimation of the total VCDs in the plume maximum and an underestimation of the integrated line densities that can be used for estimating emissions from $NO_2$ images. The two examples demonstrate the importance of 3D effects for several types of ground-based and airborne remote sensing when the atmosphere cannot be assumed to be horizontally homogeneous, which is typically the case in the vicinity of emission

sources or in cities.



# 1 Introduction

Ground-based, space-based and airborne remote sensing of air pollutants and greenhouse gases from scattered sunlight are increasingly used for air pollutant monitoring (e.g. Frankenberg et al., 2005; Richter et al., 2004; McPeters et al., 2015; Burrows et al., 1999; Zhou et al., 2012; Nowlan et al., 2016) and for source detection and emission estimation (e.g. Mijling et al., 2013; Martin et al., 2003; Russell et al., 2012; Krueger et al., 1995). The most commonly applied trace gas retrieval method in the ultraviolet, visible and near-infrared spectral range is the Differential Optical Absorption Spectroscopy (DOAS) (Platt and Stutz, 2008), which fits absorption cross sections of a trace gas to the measured spectra. The result of the DOAS analysis is a slant column density (SCD), which is the integrated trace gas concentration along the optical path of the sunlight scattered towards the spectrometer. The optical path depends on the illumination and viewing geometry, on absorption and scattering by air molecules, aerosols and clouds, and on surface reflectance.

A physically more meaningful quantity that is independent of the measurement geometry is the vertical column density (VCD), which is the integrated trace gas concentration from the ground to the top-of-the-atmosphere. The ratio between SCD and VCD is called air mass factor (AMF) (Solomon et al., 1987), which can be computed with a radiative transfer model (RTM). To account for the vertical variability of atmospheric properties, AMFs are computed for discrete vertical layers (layer AMFs) assuming horizontal homogeneity (Palmer et al., 2001; Wagner et al., 2007; Rozanov and Rozanov, 2010). In the past decades, numerous RTMs have been developed with the possibility to calculate one-dimensional layer AMFs (e.g. Berk et al., 1999; Postylyakov, 2004; Rozanov et al., 2005; Wagner et al., 2007; Spurr et al., 2001). The computation of layer AMFs is implemented in most trace gas retrieval algorithms for satellite and ground-based observations applied today (Boersma et al., 2011; Irie et al., 2011; Wenig et al., 2008; Wu et al., 2013). An alternative method is direct fitting, which is used in few algorithms (e.g. Lerot et al., 2010).

Layer AMFs assume horizontal homogeneity, which is not valid when the parameters affecting scattering and absorption along the path of the photons vary also horizontally, for example, in limb geometry near the polar vortex (Puķīte et al., 2010) or in the presence of clouds (Mayer and Kylling, 2005). Horizontal homogeneity is usually a valid assumption in coarse resolution trace gas remote sensing from satellites, where small-scale horizontal variability is averaged over a large pixel size. It is however often not valid for ground-based or airborne trace gas remote sensing in polluted environments such as cities (e.g. Hendrick et al., 2014; Popp et al., 2012; Schönhardt et al., 2015; Tack et al., 2017). This is particularly true for nitrogen dioxide ($NO_2$), which has high spatial and temporal variability due to its short lifetime (Schaub et al., 2007). Other parameters affecting the path of the measured photons like surface reflectance and aerosol distributions may also have high spatial variability in cities.

To account for horizontal inhomogeneity, one-dimensional (1D) layer AMFs need to be extended to three-dimensional (3D) box AMFs. 3D-box AMFs can be implemented most easily in radiative transfer models that compute the paths of many photons using a Monte Carlo approach to solve the radiative transfer equation (Deutschmann et al., 2011). In this study, we implemented both 1D-layer and 3D-box AMFs in the MYSTIC solver of the libRadtran RTM (Mayer and Kylling, 2005; Emde et al., 2016). The implementation was evaluated against the results of a RTM comparison study (Wagner et al., 2007). Finally,




the advantage and necessity of using 3D-box AMFs is demonstrated for a range of realistic ground-based and airborne remote
sensing scenarios.

## 2  Methods

### 2.1  Air mass factors

Atmospheric trace gases can be measured with ground-, aircraft- and space-based spectrometers that measure solar irradiance
scattered into the line of sight of the instrument (see Fig. 1). In case of aircraft- and space-based observations, a large fraction
of the measured photons usually travels along a main path (thick dashed line) representing a single reflection at the surface. In
case of ground-based observations, the measured photons must follow a path with at least a single atmospheric scattering into
the line of sight of the instrument (except for direct sun observations). Atmospheric scattering and absorption is determined
by the distribution of molecules, aerosols and clouds, and depends on the wavelength of the radiation. Molecular scattering is
particularly important in the UV range of the spectrum. Photons are absorbed by the trace gases along the optical path from
the sun to the instrument. For a weak absorber such as $NO_2$, the abundance of the trace gas along the mean optical path can be
obtained by fitting an absorption cross section to the measured spectrum. Thereby, the mean optical path is the total length of
all individual photon paths divided by the number of photons collected by the instrument. The result of the fit is a slant column
density (SCD), which is defined as

$$SCD = \int_{path} c(l)dl \tag{1}$$

with trace gas concentration $c$ and optical path $l$. SCDs are not an intrinsic property of the atmosphere since they depend on the
illumination and viewing geometry. Therefore, for most applications, the main quantity of interest is the vertical column density
(VCD), which is the integrated amount of the trace gas in a vertical column from the surface to the top of the atmosphere. It is
defined as

$$VCD = \int_{z_0}^{TOA} c(z)dz \tag{2}$$

with surface elevation $z_0$ and top-of-the-atmosphere $TOA$. The ratio between SCD and VCD and is called air mass factor, thus

$$AMF = \frac{SCD}{VCD}. \tag{3}$$

AMFs can be computed for a vertically varying atmosphere by dividing the atmosphere in layers with uniform properties
(see Fig. 1a and 1b). The total AMF is then computed from the individual layer AMFs as

$$AMF = \frac{\sum_{k=1}^{n_z} AMF_k VCD_k}{\sum_{k=1}^{n_z} VCD_k} \tag{4}$$





**Figure 1.** Illustration of the difference between 1D-layer (upper panels) and 3D-box AMFs (lower panels) for two scenarios with downward-looking spaceborne (left) and upward-looking ground-based (right) observations. Selected photon paths are shown as dashed lines. 1D-layer AMFs implicitly assume horizontally uniform atmospheric and surface properties, whereas 3D-box AMFs fully account for both vertical and horizontal variability.

with $AMF_k$ and $VCD_k$ being the AMF and VCD in the $k$-th layer, respectively. The total AMF is thus not only a function of the atmospheric properties in each layer but also of the shape of the vertical profile of the trace gas (Palmer et al., 2001).



Similarly, the atmosphere can be divided in boxes in all three dimensions $(i, j, k)$ with homogeneous optical properties for each box (see Fig. 1c and 1d). The total AMF can be computed from the 3D-box AMFs $AMF_{i,j,k}$ as

$$AMF = \frac{\sum_{i=1}^{n_x} \sum_{j=1}^{n_y} \sum_{k=1}^{n_z} AMF_{i,j,k} VCD_{i,j,k}}{\sum_{k=1}^{n_z} VCD_k} \tag{5}$$

where the denominator is a sum over VCDs in $k$ different vertical layers that could, for example, be taken at the location of an instrument or above the ground pixel of a aircraft- or space-based instrument. In this case, the AMF can be interpreted as the instrument sensitivity to $NO_2$ for measuring that specific VCD. Notice that in previous studies (Rozanov and Rozanov, 2010, e.g.) 1D-layer AMF were some times referred to as box AMFs. In this study, we will use the terms 1D-layer and 3D-box AMFs to clearly distinguish between them.

## 2.2 Implementation of AMFs in MYSTIC

The libRadtran RTM (available at www.libradtran.org) can be used to calculate basic radiative quantities with different numerical solvers (Mayer and Kylling, 2005; Emde et al., 2016). One of its solvers is MYSTIC, which uses the Monte Carlo technique to trace individual photons on their way from the source (e.g. sun) to the target (e.g. measurement instrument). Scattering, absorption and reflection processes are treated as random decisions with respective probability distributions. MYSTIC calculates radiative quantities (irradiance, actinic flux at levels, radiance, absorption, emission, actinic flux, photon's path length and air mass factors) in 1D or 3D domains in spherical geometry or in plane parallel geometry (Emde and Mayer, 2007; Emde et al., 2017). 1D-layer and 3D-box AMFs were implemented following the same methodology as in McArtim, which to our knowledge is the only other existing RTM capable of computing 3D-box AMFs (Deutschmann et al., 2011; Richter et al., 2013). Note that McArtim is no longer actively developed.

AMFs depend on absorption and scattering processes affecting the light path in the atmosphere. AMFs can be readily calculated from the photon paths simulated by a Monte Carlo radiative transfer model. The Monte Carlo technique traces the paths of individual photons by describing the effects of absorption, scattering and reflection as random events with specific probabilities (Mayer, 2009). To obtain a robust measure of the mean optical path, a large number of photon paths needs to be traced.

SCDs, VCDs and AMFs can be computed for the whole atmosphere, for individual vertical layers, or for individual 3D boxes. For the general case of an atmospheric box $i$ with constant concentration and optical properties, the AMF can be written as:

$$AMF_i = \frac{SCD_i}{VCD_i} = \frac{\int_{path} c_i dl}{\int_{z_i}^{z_{i+1}} c_i dz} = \frac{\int_{path} dl}{h_i} = \frac{L_i}{h_i} \tag{6}$$

where $L_i = \int_{path} dl$ is the mean optical path within the box of all photons that reach the instrument and $h_i$ is the height of the box. Since the 3D-box/1D-layer AMFs are usually simulated for a sensor at a specific location in a three-dimensional model-domain, the photons are traced backwards from the sensor towards the sun to increase computationally efficiency as described in Marchuk et al. (1976) and Emde and Mayer (2007). In addition, the commonly used variance reduction method, known as



"local estimate", is applied at each scattering event (Marshak and Davis, 2005). The method computes the probability of an

individual photon to be scattered into the direction of the sun that is assigned as a weight $w_n$ to the photon. The weights of all photons can be summed up to obtain the radiance at the sensor. When a photon is scattered, a weighted photon path-length $(w_n \cdot l_i)$ is also calculated, where $l_i$ are the path-lengths in each individual box $i$ traversed by the photon before the scattering event. The mean optical path within a box $i$ is then obtained by summing up the weighted photon path-lengths of all photons:

$$L_i = \frac{\sum_n^N w_n l_{i,n}}{\sum_n^N w_n} \tag{7}$$

where $N$ is the total number of photons. $L_i$ is then divided by the height of the box/layer to obtain the 3D-box/1D-layer AMF.

## 3 Validation of the AMF modules

### 3.1 Evaluation scenarios

The implementation of the 1D-layer and 3D-box AMF module in MYSTIC was evaluated against the results of different RTMs presented in an extensive RTM comparison study (Wagner et al., 2007). The simulated scenarios are representative for ground-

based Multi-Axis-DOAS (MAX-DOAS) measurements of scattered sunlight spectra for different elevation angles (see Figure 1b and 1d for the case of zenith-sky observations). The nine models included four models using full spherical geometry, four models using spherical geometry only for a subset of interactions, and one model using plane parallel geometry. The 1D-layer AMFs computed by these models agreed very well with differences mostly below 5%, which could mainly be attributed to the different treatments and approximations of the Earth's sphericity and to model initialization parameters (Wagner et al., 2007).

For the comparison, we computed 1D-layer and 3D-box AMFs with MYSTIC in plane parallel geometry as well as 1D-layer AMFs in spherical geometry for all scenarios presented in Wagner et al. (2007). 3D-box AMFs have not yet been implemented with spherical geometry.

1D-layer and 3D-box AMFs were computed for five wavelengths (310 nm, 360 nm, 440 nm, 477 nm, 577nm), seven elevation angles (1°, 2°, 3°, 6°, 10°, 20°, 90°), and three aerosol scenarios (aerosol extinction of 0.0, 0.1 and 0.5 km$^{-1}$). For the

aerosol scenarios, an aerosol layer was prescribed between 0 and 2 km with an asymmetry parameter of 0.68 and a single scattering albedo of 1.0. No aerosols were prescribed above 2 km. For the simulations 17 vertical layers were used with 100 m below 1000 m and mostly 1000 m resolution above (see Table 1 in Wagner et al., 2007). Profiles of temperature, pressure, density and ozone concentration were taken from the US standard atmosphere (United States Committee on Extension to the Standard Atmosphere, 1976). Ozone cross sections (in cm$^2$) were $9.59 \times 10^{-20}$, $6.19 \times 10^{-23}$, $1.36 \times 10^{-22}$, $5.60 \times 10^{-22}$ and

$4.87 \times 10^{-21}$ at 310 nm, 360 nm, 440 nm, 477 nm and 577 nm, respectively. Other atmospheric absorbers were ignored. Further details can be found in Wagner et al. (2007). For each scenario, we traced 1 million photons, which balances statistical noise expected from a Monte Carlo approach with computation time. The computed 3D-box AMFs were integrated horizontally to obtain 1D-layer AMF that can be compared with the 1D-layer AMFs from other models. MYSTIC was mainly compared to SCIATRAN (Version 2.2, Rozanov et al. (2005)). SCIATRAN was chosen because it agrees well with the mean of the models in



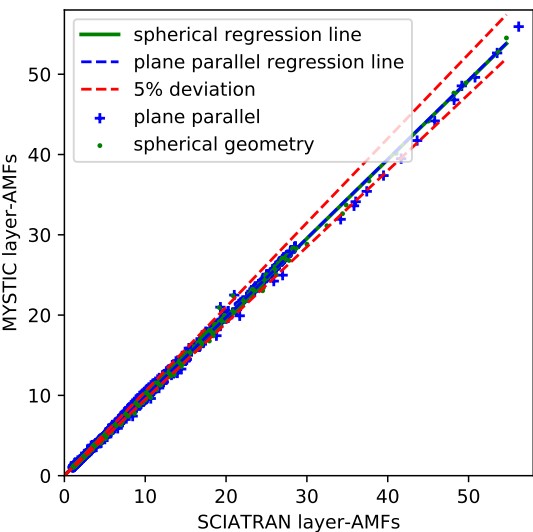

**Figure 2.** Scatter plot of MYSTIC 1D-layer AMFs computed with spherical (green dots) and plane parallel geometries (blue cross) against 1D-layer AMFs computed with SCIATRAN plane parallel and spherical for 67 scenarios with 17 layers (1139 points). The solid green line is the regression fit to the spherical geometry points and the dashed blue line is the regression to fit the plane parallel geometry points.

Wagner et al. (2007), and also because it is based on the discrete ordinate method to solve the radiative transfer equation, which is fundamentally different from a Monte Carlo solver, and finally because it offers both plane parallel and spherical solutions. In addition, we compared MYSTIC to the mean of eight of nine RTMs in the comparison study. The PROMSAR/Italy model was not included because of its large deviation from the mean (see Wagner et al., 2007, for details).

## 3.2 Validation results

The comparison of 1D-layer AMF profiles calculated with the MYSTIC 1D modules with SCIATRAN for the 67 observation scenarios used in Wagner et al. (2007) is summarized in Fig. 2 in the form of a scatter plot. The horizontally integrated AMFs from MYSTIC's 3D module perfectly agree with its 1D module with plane parallel geometry within the statistical noise of the Monte Carlo approach. When tracing 1 million photons, the difference between 1D and 3D module was smaller than 0.5%. Therefore, only results from the 1D module were plotted against the SCIATRAN results. The agreement between MYSTIC

and SCIATRAN is very good for almost all scenarios with relative differences mostly below 5%. 97% of the compared points are within a relative difference of 5% for spherical geometry and 92% for plane parallel geometry. The mean of the relative differences for spherical geometry is 0.9% and its standard deviation 2.0% and for the plane parallel geometry the mean is 0.3% with a standard deviation of 2.7%.





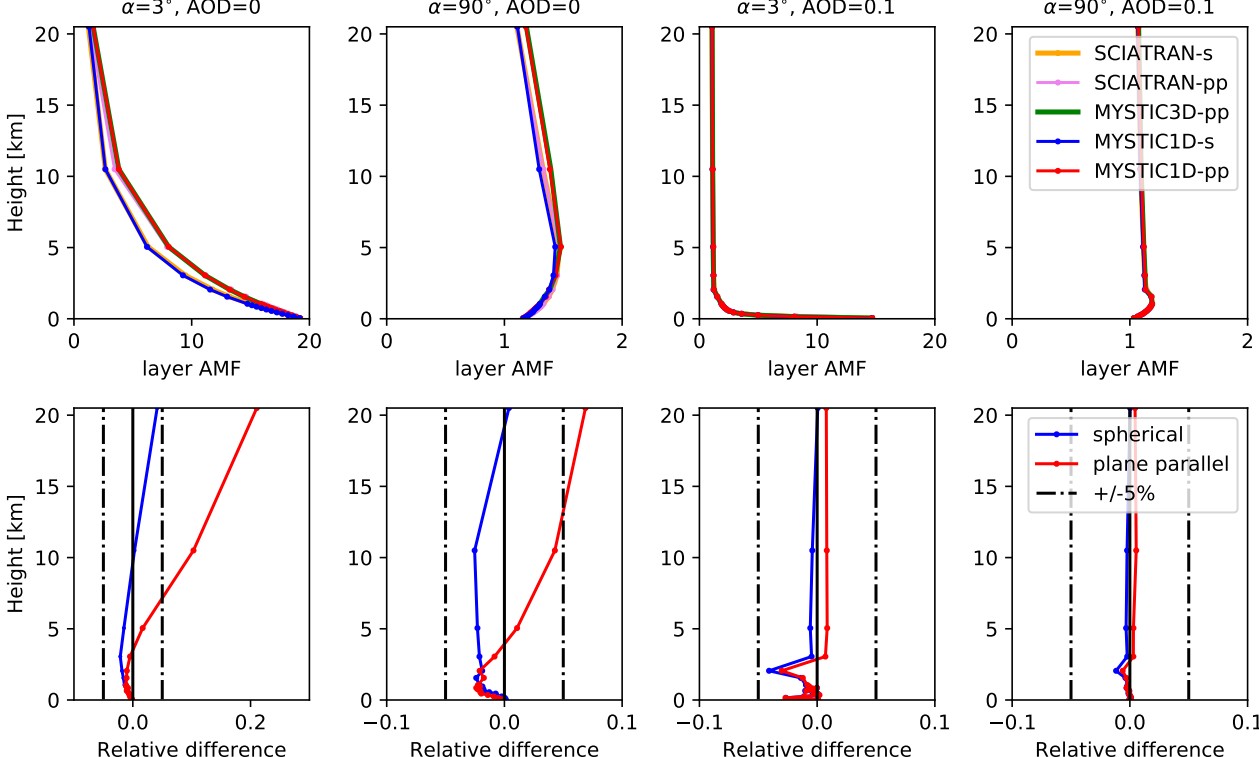

**Figure 3.** Upper row: AMF profiles for MYSTIC 1D spherical geometry (blue), 1D plane parallel geometry (red) and 3D plane parallel geometry (green) for two selected elevation angles of 3° and 90°, a SZA of 20°, with and without aerosol for radiation at 577 nm. Corresponding profiles computed with the SCIATRAN RTM are shown for comparison. Lower row: Profile of relative differences of MYSTIC and SCIATRAN results in spherical (blue) and plane parallel geometry (red) (Wagner et al., 2007).

To illustrate the differences in AMF profiles between the two RTMs, we selected four scenarios with a wavelength of 577 nm
because at this wavelength we observe comparatively large differences between the two models. To illustrate an usual scenario with low difference, we also selected the same scenarios but with a 360 nm wavelength. The upper row of Figures 3 and 4 show MYSTIC 1D-layer AMF profiles for the selected scenarios with a low elevation angle of 3° and a high elevation angle of 90° (zenith) without and with aerosols, respectively. For comparison, the corresponding profiles computed with SCIATRAN are also shown. The lower row presents the relative differences between MYSTIC and SCIATRAN. Since plane parallel and
spherical modes have different geometrical assumptions, we compare plane parallel models and spherical models separately.

In the upper atmosphere, the 1D-layer AMFs decrease with altitude in all scenarios (Fig. 3 and Fig. 4), because the atmospheric density is decreasing, which lowers the amount of scattering and, correspondingly, the mean photon path length. In the lowest layers, however, the profile shapes are different for the two elevation angles with a rapid decrease with altitude in the low elevation scenarios and a local maximum between 2 and 5 km in the high elevation scenarios. This local maximum is

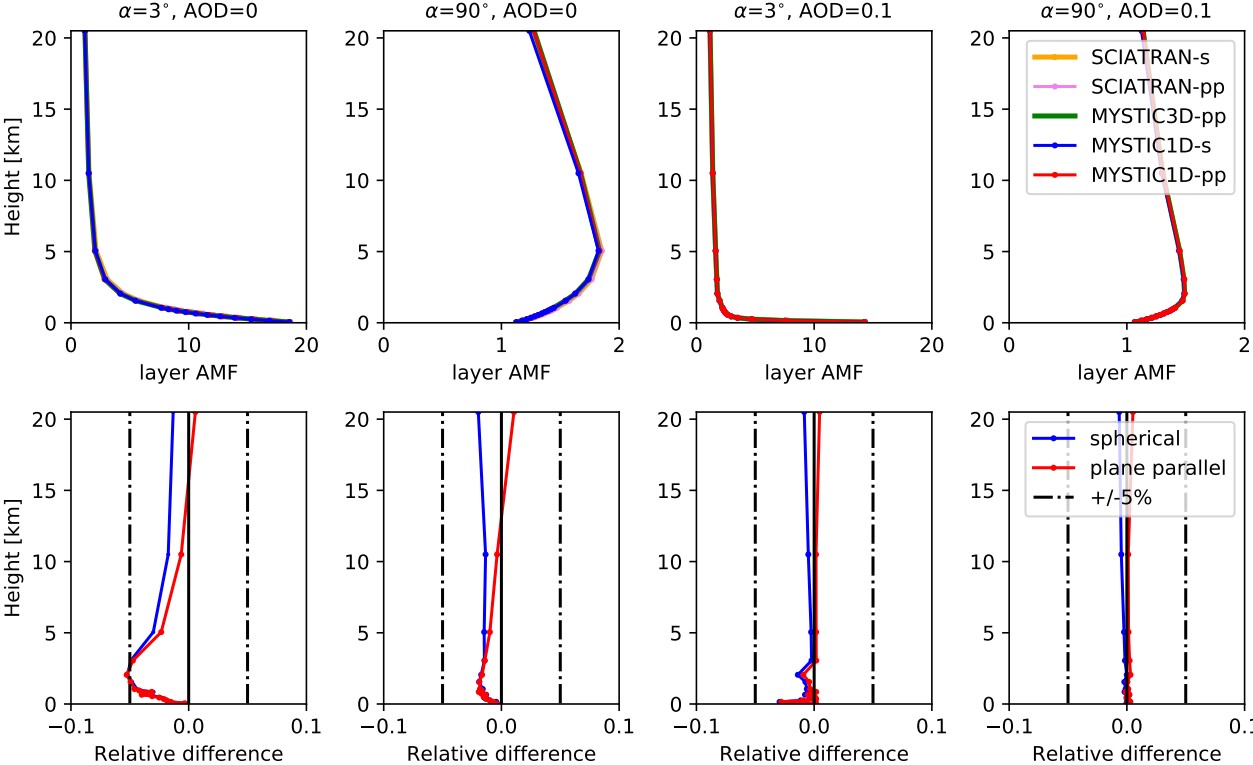

**Figure 4.** Upper row: AMF profiles for MYSTIC 1D spherical geometry (blue), 1D plane parallel geometry (red) and 3D plane parallel geometry (green) for two selected elevation angles of 3° and 90°, a SZA of 20°, with and without aerosol for radiation at 360 nm. Corresponding profiles computed with the SCIATRAN RTM are shown for comparison. Lower row: Profile of relative differences of MYSTIC and SCIATRAN results in spherical (blue) and plane parallel geometry (red) (Wagner et al., 2007).

caused by multiple scattering, which contributes to the horizontal light paths in those layers. The reduction towards the surface in the latter scenarios is due to the low surface albedo. For an elevation angle of 3°, AMFs are high close to the ground because of the long light path in the layers due to the low elevation angle. Since aerosols increase scattering, photon path lengths and correspondingly 1D-layer AMFs are low in the lowest 2 km, when an aerosol layer is present.

1D-layer AMFs computed with spherical and plane parallel geometry show noticeable differences for long wavelengths and low aerosol extinction, especially at altitudes above 5 km where extinction coefficients are small (see upper and lower left part in Fig. 3). In plane parallel geometry, if one of these photons is travelling horizontally, it will strongly contribute to increase the mean photon path in that specific layer. In spherical mode, the same photon would change layer because of the curved atmospheric layers and therefore its contribution to the mean photon path will be divided between the crossed layers. Furthermore, in a curved atmosphere, the zenith angle of the photon, that was initially travelling horizontally, will increase. At





low altitude these effects are smaller and, conversely, 1D-layer AMFs computed with spherical and plane parallel geometry agree better (mostly < 5%).

AMF profiles calculated with MYSTIC generally agree very well with those calculated with SCIATRAN with relative differences mostly smaller than 5%. However, significant differences (up to 23% relative difference) are seen between the plane parallel solutions of the two models above 5 km for the scenarios without aerosols at 577 nm (Figure 3). In contrast to

the plane parallel case, the spherical solution of MYSTIC is in good agreement with the spherical solution of SCIATRAN. The difference between SCIATRAN plane parallel and MYSTIC plane parallel is attributed to the different solution methods of the radiative transfer equation. A possible explanation is the following: In discrete ordinate methods, the directions of the radiation field are discretized and do not include the exact horizontal direction, for which in plane parallel geometry the photon path-length becomes extremely large in an optically thin medium like the higher atmosphere. In a Monte Carlo model, this

horizontal direction is included, therefore the 1D-layer AMF might be larger. This hypothesis could be tested by including more streams (discrete directions) in SCIATRAN and verifying if it approaches the higher AMFs from the MYSTIC solution.

The simulations for the same scenarios but with 360 nm wavelength agree very well with SCIATRAN for both spherical and plane parallel geometries (relative difference <5%). The differences mentioned above are much smaller at this wavelength because atmospheric scattering events increase with lower wavelength and thus, prevent those very long photon paths. We also

investigated a scenario with a wavelength of 440 nm, which is a typical wavelength of the window used for $NO_2$ fitting (see figure S4 in the Supplement), for which MYSTIC and SCIATRAN also agree very well (< 5% relative difference), but as for simulations at 577 nm discussed above, the simulations at 440 nm show significant differences between plane parallel and spherical geometry for layers above 5 km. These differences are, however, smaller than at 577 nm because the optical thickness of Rayleigh scattering is higher at 440 nm.

Overall, MYSTIC agrees very well with SCIATRAN with differences mainly smaller than 5%. An exception is the high elevation scenario without aerosols, where the plane parallel solutions of MYSTIC and SCIATRAN differ by up to 23% for a wavelength of 577 nm at altitudes above 5 km. It should be noted that for these cases the 1D-layer AMFs are very small and therefore the absolute differences, which are relevant for most applications, are also small. 1D-layer AMFs computed with MYSTIC also agree very well with the other models presented in Wagner et al. (2007). Differences larger than 5% are mainly

attributable to differences between plane parallel and spherical solutions (see Supplement). When comparing MYSTIC with the mean of the models, 88.3% of the compared points are within a relative difference of 5% for spherical geometry, 81.5% for plane parallel geometry, and 97.5% for the mean of plane parallel and spherical geometry. The mean of spherical and plane parallel geometry agrees best because the models in Wagner et al. (2007) represents a mixture of spherical and plane parallel solutions.

## 4 3D-box AMFs for MAX-DOAS observations

MAX-DOAS is a ground-based passive remote sensing technique allowing to retrieve vertical concentration profiles of trace gases and aerosols (Wagner et al., 2004; Frieß et al., 2006; Irie et al., 2011; Hönninger and Platt, 2002). Information on the





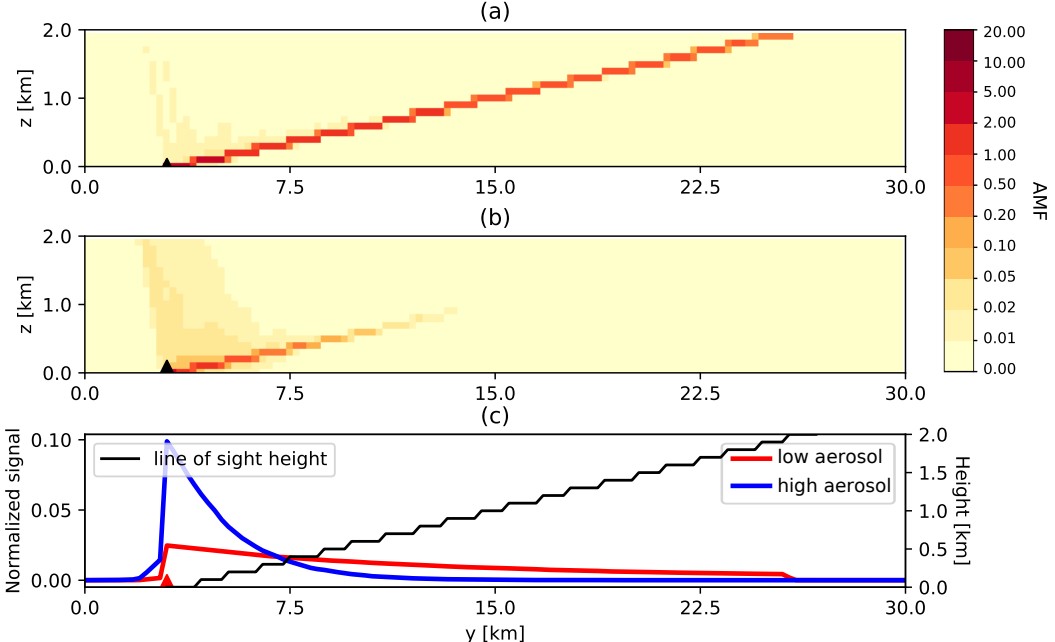

**Figure 5.** Cross section of 3D-box AMFs for a MAX-DOAS scenario with an instrument (black triangle) at the ground (z=0 km, x=20 km, y=3 km) pointing northwards and slightly upwards at a viewing angle of 5°. The sun is at an azimuth angle of 344.7° and a zenith angle of 24.6°. The relative azimuth angle between sun and viewing direction is 164.7°. AMFs were simulated with two aerosol scenarios: a rural type aerosol representative of spring-summer conditions in the aerosol layer (0-2 km), with a visibility of 50 km (a) and a visibility of 10 km (b) and a background aerosol above 2 km. Decay of vertically integrated AMFs with distance to the instrument (c) for the same scenarios with standard (red) and high aerosols (blue) as in (a) and (b). The altitude of the line of sight as a function of distance is shown in black.

vertical distribution is obtained by measuring spectra for a prescribed sequence of elevation angles. Observations at different elevation angles have different sensitivity to the concentration in a given vertical layer. 3D-box AMFs as computed by MYSTIC

are particularly suitable to illustrate this, because 3D-box AMFs are a direct representation of the spatial distribution of the sensitivity of the measurements.

To illustrate the 3D distribution of 3D-box AMFs for a typical MAX-DOAS measurement, we simulated 3D-box AMFs at 450 nm for two scenarios with low and high aerosol optical depth, which correspond to a visibility of 50 and 10 km in the atmospheric boundary layer, respectively. 450 nm is a typical wavelength for light absorption by $NO_2$. The instrument points

northwards with an azimuth angle of 180° and an elevation angle of 5°. The solar azimuth angle is 344.7° (164.7° relative azimuth angle) and the solar zenith angle is 24.6°. The MYSTIC input file is provided in the Supplement.

Figures 5a and 5b show the 3D-box AMFs in the plane of the line of sight of the instrument for the two scenarios. In both cases, 3D-box AMFs are highest along the line of sight and reduce with distance from the instrument. Most of the photons collected by the instrument experienced a single scattering into the line of sight of the instrument. With increased aerosol



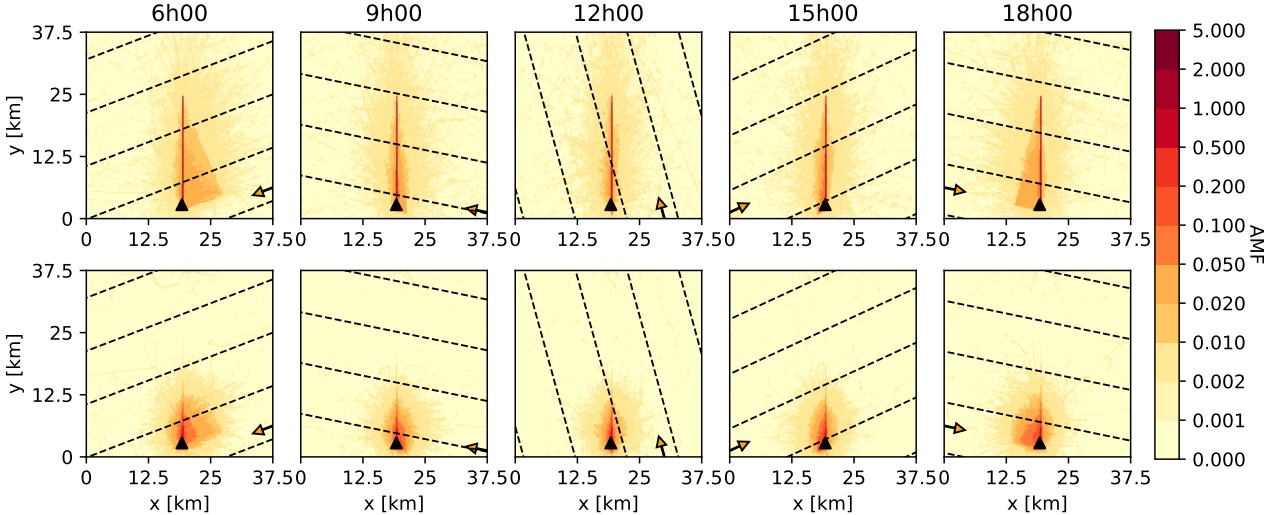

**Figure 6.** Top: Vertically integrated 3D-box AMFs in the boundary layer (z < 2.0 km) for an instrument at the ground pointing northwards with an instrument zenith angle of 5° for different times of the day on a 21st of June in Zurich. Solar zenith angles are 77.4°, 47.5°, 24.6°, 38.6° and 68.5° and solar azimuth angles are 249.0°, 281.5°, 344.7°, 65.2° and 101.7°. The arrows point away from the sun and the dashed lines show the direction of photons coming from the sun. AMFs were simulated with a rural type aerosol representative of spring-summer conditions in the aerosol layer (0-2 km) with a visibility of 50 km and a background aerosol above 2 km. Bottom: Same as above but for a scenario with increased aerosol (visibility of 10 km).

amount (visibility of 10 km), photons scattered into the line of sight far away from the instrument have a high chance of being scattered out again. As a result, the sensitivity rapidly (within a few kilometers) decreases along the line of sight with increasing distance from the instrument. Multiple scattering becomes more important in this scenario, which explains the enhanced sensitivity to layers below and above the line of sight within a distance of up to 4 km of the instrument. The decrease of AMF with distance is further illustrated for the two scenarios in Figure 5c which shows the vertically integrated AMFs (in
the aerosol layer) as a function of distance $y$ to the instrument normalized with AMFs integrated horizontally in $y$ direction. The figure also shows the height of the main optical path as a function of $y$.

To illustrate the horizontal spread of the sensitivity of the MAX-DOAS measurements in the atmospheric boundary layer, Fig. 6 shows horizontal distributions of vertically integrated 3D-box AMFs (0-2 km) for the same scenarios with low (top row) and high (bottom row) aerosols and for five different sun positions corresponding to different times of the day on 21 July in
the city of Zurich. The horizontal distribution of AMFs shows high values along the line of sight of the instrument but also in a surrounding region, which is up to a few kilometers wide. This region is wider for larger relative azimuth angles and is inclined towards the direction of the sun. The simulations show that the MAX-DOAS measurements are not only sensitive to $NO_2$ along the line of sight but that they are also influenced by neighboring regions a few kilometers away.



For the different scenarios we evaluated which part of the signal originated from a 0.25 km wide region centered on the northward pointing line of sight (referred to as main line in the following), and which part crossed boxes outside this range. For the low aerosol scenario, between 63% and 70% originated from the main line. Thus, up to 37 % of the signal originated from photons crossing neighbouring boxes. For the high aerosol scenario with enhanced scattering, the part of the signal originating from the main line was correspondingly lower, between 30% and 41%. The lower values correspond to the scenarios with higher relative azimuth angles.

Depending on the viewing direction of the instrument relative to the position of nearby emission sources, this temporally varying spatial sensitivity could introduce a diurnal cycle in the measurement even when the trace gas concentration field was constant in time. Understanding the horizontal distribution of the sensitivity to $NO_2$ and its variation in time is thus particularly important for the interpretation of MAX-DOAS observations in polluted regions like cities with strong $NO_2$ gradients, for which 3D-box AMFs can be a valuable tool.

## 5 3D-box AMFs for airborne observations

In this section, we demonstrate the effect of the spatial variability of 3D-box AMFs on airborne $NO_2$ imaging spectroscopy. For this purpose, we simulated a $NO_2$ plume emitted from a stack to generate a scenario with a distinct three-dimensional trace gas structure. An airborne spectrometer was then assumed to fly parallel to the plume axis and to sample the plume in across-track direction (see dashed line in Figure 8). We illustrate the distinct 3D-structure of the sensitivity of the measurements to $NO_2$ (as represented by the 3D-box AMFs) and demonstrate the limitations of using 1D-layer AMFs for such observations.

### 5.1 Synthetic observations of a $NO_2$ stack emission plume

The $NO_2$ plume was computed with the Graz Lagrangian Model (GRAL) (Oettl, 2015) for a 262.5 m tall stack located at x=1.9 km and y=1.3 km. $NO_2$ molecules were released at this altitude at a constant rate of $40\,kg\,h^{-1}$. $NO_x$ chemistry was ignored for simplicity. The model domain had a size of 4 km × 4 km and extended from the surface to 21 km altitude. The simulated $NO_2$ was sampled on an output grid with a 100 m horizontal resolution and 20 vertical levels with 25 m resolution from 0 to 500 m. For the simulation we assumed neutral atmospheric stability and southerly wind with a speed of $5\,m\,s^{-1}$ at 12 m above ground. The full vertical wind profile is generated within the model based on similarity theory. The $NO_2$ background from the US Standard Atmosphere (United States Committee on Extension to the Standard Atmosphere, 1976) was added to the simulated $NO_2$ field, that was extended to 21 km altitude (see vertical resolution profile in Supplement). The resulting $NO_2$ VCDs are shown in Figure 8a. In the following, the simulated $NO_2$ concentration field and the corresponding $NO_2$ VCDs are referred to as the true $NO_2$ field and as the true total VCD, respectively. The true VCD will be used as a reference to demonstrate the limitations of 1D radiative transfer calculations.

Using MYSTIC, we computed the SCDs that would be observed from an airborne push-broom spectrometer flying parallel to the plume axis from south to north at an altitude of 6 km. The field of view in across-track direction of the instrument covers the full $x$-direction of the model domain. The SCDs were obtained by computing 3D-box AMFs for each single observation



**Table 1.** MYSTIC input parameters for the emission stack scenario.

| Parameter | Value |
| --- | --- |
| Wavelength [nm] | 460 |
| Solar zenith angle [°] | 0, 40, 20, 60 |
| Solar azimuth angle [°] | 90, 0, 180, 270 |
| Viewing zenith angle [°] | 0 to 26.6 |
| Viewing azimuth angle [°] | 90 / 270 |
| Surface albedo | 0.2 |
| Aircraft position x [km] | 2.9 |
| Aircraft position y [km] | 0-4 |
| Aircraft position z [km] | 6 |
| Domain size [boxes] | $40 \times 40 \times 47$ |
| Horizontal resolution [m] | 100.0 |
| Vertical resolution (0 - 7km) [m] | 25 |

(i.e. for each ground pixel) and multiplying these AMFs with the 3D $NO_2$ field from the simulation (which corresponds to the nominator in Eq. 5).

As an example, Fig. 7 illustrates the 3D-box AMFs for an instrument pointing downwards at a zenith angle of 4.8° and an azimuth angle of 90°. The sun is placed in the west (SAA=90°) at a SZA of 20°, i.e. the instrument is facing the sun. The figure shows the 2D cross-section of 3D-box AMFs in the principal plane of the observations which aligns with the $x$-$z$ plane in this geometry. The panels to the right and below the main figure show vertically and horizontally integrated 3D-box AMFs, i.e. column and layer AMFs, respectively. The layer AMFs are identical to 1D-layer AMFs. The 3D-box AMFs are high along the line of sight of the instrument and largest just below the aircraft. Most photons travel directly along the geometric path from the sun to the ground pixel and then to the instrument. Although 3D-box AMFs are highest along the geometric path due to the relatively bright surface, a non-negligible fraction of photons is scattered into the line of sight without reaching the surface leading to an increase in 3D-box AMFs within a parallelogram bounded by the line of sight and the position of the sun.

The column AMFs (lower panel) are highest close to the instrument and decrease with distance to the instrument in $x$-direction due to atmospheric scattering. After the "reflection point", values continue to decrease with distance to the instrument but at a lower rate. Due to periodic boundaries this decrease continues on the right of the instrument ($x \geq 3.9$ km). Layer AMFs (i.e. 1D-layer AMFs) (right panel) are highest directly below the instrument. They change by a factor of two at the altitude of the aircraft because layers below are crossed (at least) twice by the photons, while layers above are only crossed once.

3D-box AMFs and corresponding SCDs were computed for four different solar zenith angles and four different relative azimuth angles between the sun and the plume axis (and flight direction). We used a default aerosol scenario with a rural type aerosol representative for spring-summer conditions in the planet boundary layer (PBL)(0-2 km) and a background aerosol above 2 km (visibility of 50 km in the PBL). The parameters used for the AMF calculation are summarized in Tab. 1. Note that

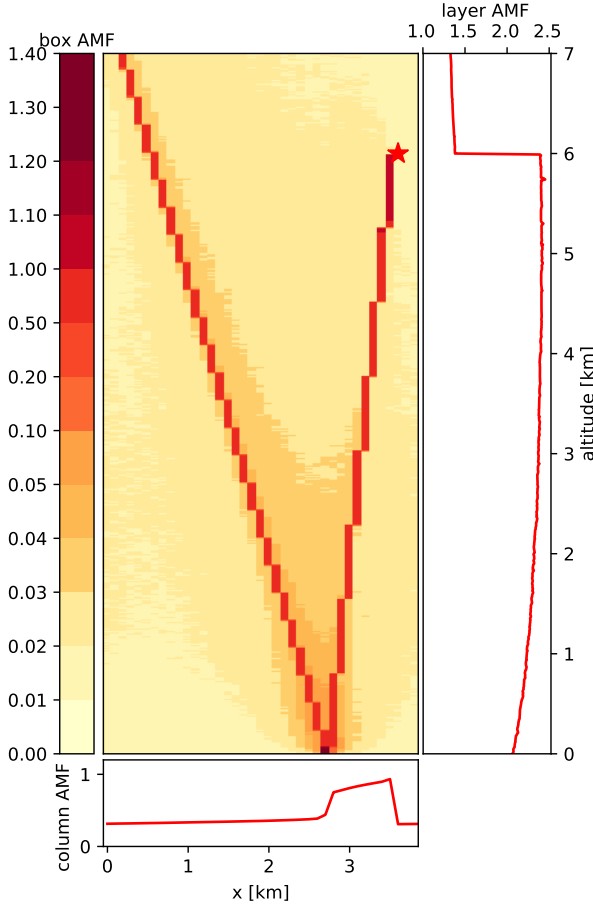

**Figure 7.** 3D-box AMFs cross section at y=1.4 km for the aircraft scenario presented in this section. Aircraft (red star) placed at z=6 km, x=2.9 km and y=1.4 km pointing eastwards. The sun is at SAA = 90° (west) with a SZA of 20°. Right: Vertical profile of horizontally integrated AMFs (1D-layer AMFs). Bottom: Horizontal profile of vertically integrated AMFs (column AMFs). The default properties are a rural type aerosol in the boundary layer, background aerosol above 2 km, spring-summer conditions and a visibility of 50 km.

with perfect knowledge of the relative $NO_2$ distribution, the true total VCD could be reproduced exactly from the SCDs using 3D radiative transfer calculations.

The SCDs computed for the scenario with the sun illuminating the scene from the west at a solar zenith angle of 40° are presented in Fig. 8b. The SCCs are larger than the VCDs (panel a) because the AMFs (panel c) are generally larger than 1. The
SCD plume is wider and shifted towards the east compared to the VCD plume. The widening is due to both geometric effects and atmospheric scattering. Geometric effects are caused by the fact that photons following the main geometric path from the sun to the surface and to the instrument may traverse the plume either on the way from the sun to the surface or from the surface to the instrument (or both). These two pathways are separated horizontally. For high solar zenith angles (here SZA=40°) this





leads to two SCD maxima close to the source as seen in Fig. 8b. The westerly maximum corresponds to the direct observation
of the plume (photons reflected by the surface pass the plume on the direct way to the aircraft) whereas the easterly maximum
corresponds to its mirror image (photons first travel through the plume before they get reflected at the surface and reflected
to the aircraft). This is further illustrated in Fig. 9, where 2 of the 3 illustrated direct paths (i.e. 3 instrument zenith angles)
cross the $NO_2$ maximum (main photon path (1) and (3) in Fig. 9). The main photon path for the observation angle (2) in
Fig. 9 misses the plume maximum, which is why total SCD is lower for this observation. Atmospheric scattering leads to an
additional horizontal smoothing of the plume, but in the case of a medium high surface albedo of 0.2 the geometric effects
dominate.

## 5.2    Limitations of VCDs calculated from 1D-layer AMFs

For each scenario, total AMFs were also computed from 1D-layer AMFs, which requires a $NO_2$ profile (Eq. 4). The most
obvious approach is to use the true $NO_2$ profile above the ground pixel the instrument is pointing towards, which is based
on the idea that the AMF is used to convert an SCD to a VCD above a ground pixel (Fig 8 2nd row). Alternatively, a $NO_2$
background profile from the US Standard Atmosphere (United States Committee on Extension to the Standard Atmosphere,
1976) was used for each ground pixel, which assumes that no information on the spatial variability of $NO_2$ is available (Fig 8
3rd row).

Figures 8f and 8i show the total AMFs computed with the true and background $NO_2$ profile, respectively. In both cases,
AMFs increase with distance from the aircraft due to the increasing viewing zenith angle. For the true $NO_2$ profiles, AMFs are
higher inside the plume. This can be explained by the fact that the measurements are more sensitive to $NO_2$ inside the plume
than to the background $NO_2$ outside because the plume is located at an altitude where the 1D-layer AMFs are higher.

Figures 8d and 8g show the VCDs obtained by dividing the true SCDs in Fig. 8b by the 1D-layer AMFs in Fig. 8f and
8i, respectively. Since geometric distortions and horizontal smoothing due to scattering cannot be corrected for when using a
1D radiative transfer model, all structures seen in the SCDs are essentially preserved in the VCDs including the double peak
structure, the widening of the plume, and the horizontal displacement. Figures 8e and h show the differences of these VCDs
from the true VCDs. In both cases, the location of the plume is shifted towards the aircraft relative to the true position. Within
the maximum of the plume, this displacement leads to an underestimation of the true VCDs by -60.8 $\mu$mol/m$^2$ when using the
$NO_2$ profile (8e) above the ground pixel and by -54.6 $\mu$mol/m$^2$ when using the constant $NO_2$ profile (Figure 8h).
The displacement of the calculated VCD plume and the magnitude of the bias depend on the position of the sun as demon-
strated in Fig. 10 and Fig. 11. The shift increases with increasing SZA due to the geometric effects explained earlier. The
relative azimuth angle between the viewing direction and the sun also plays a critical role. The displacement is smaller when
the aircraft is flying directly away from the sun (SAA=0°) or towards the sun (SAA=180°) and the sun illuminates the scene
along the plume axis, but even in these cases it is not negligible. Biases are typically larger when the spatial displacement is
large.



**Figure 8.** Airborne remote sensing of an $NO_2$ plume emitted from a 262.5 m tall stack located at x=19 km and y=13 km. The aircraft flies at an altitude of 6 km from south to north at $x = 2.9$ km (dashed line) parallel to the plume axis and samples the plume in across-track direction. The sun is located in the west (small arrow in panel a) at a zenith angle of 40°. The panels show (a) simulated (true) $NO_2$ VCDs, (b) synthetic SCDs computed from the simulated $NO_2$ distribution by applying 3D-box AMFs, and (c) 3D-box AMFs computed with MYSTIC. The 2nd row shows (d) VCDs calculated from the SCDs using 1D-layer AMFs and the "true" $NO_2$ profile above the ground pixel pointed by the instrument, (e) the difference between calculated and true VCDs, and (f) total AMFs from the MYSTIC 1D module. The 3rd row (g)-(i) shows the same as (d)-(f) but using the background $NO_2$ profile to compute AMFs.





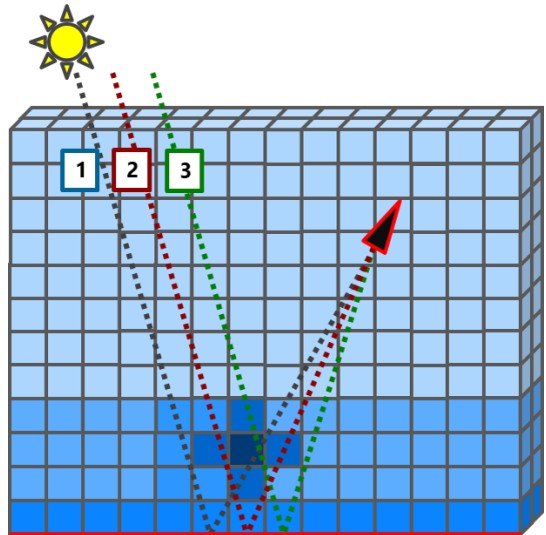

**Figure 9.** Schematic of the across-track measurement by the aircraft measuring a $NO_2$ plume (dark blue corresponding to high $NO_2$ concentrations) with 3 main photon paths for 3 measurement geometries (1, 2, 3).

**Table 2.** Estimated $NO_2$ emissions from the retrieved VCD fields obtained from 1D-layer AMFs under different solar zenith angle (SZA) and solar azimuth angles (SAA).

| Scenario | True VCD | Solar zenith angle (with SAA=90°) | | | | Solar azimuth angle (with SZA=40°) | | | |
|---|---|---|---|---|---|---|---|---|---|
| | | 0° | 20° | 40° | 60° | 0° | 90° | 180° | 270° |
| Line density $[g\,m^{-2}]$ | 1.30 | 1.18 | 1.13 | 1.13 | 1.09 | 0.82 | 1.13 | 1.06 | 1.11 |
| Flux $[kg\,h^{-1}]$ | 42.65 | 38.61 | 37.01 | 37.08 | 35.62 | 26.83 | 37.08 | 34.61 | 36.47 |
| Relative bias [%] | - | -9.48 | -13.22 | -13.06 | -16.49 | -37.09 | -13.06 | -18.86 | -14.49 |

### 5.3 Plume flux estimation

A possible application of airborne imaging spectroscopy is the estimation of $NO_2$ emissions from point sources. Measurements from airborne spectrometers have been used, for example, to estimate $CO_2$ emissions from power plants (Krings et al., 2011) or $CH_4$ emissions from coal mine ventilation shafts (Krings et al., 2013). The emissions can be estimated using a mass balance approach by integrating the $NO_2$ VCD enhancement above the background across the plume and multiplying this integral (referred to as line density in the following) with a mean wind speed to obtain a flux. The flux is equivalent to the source strength under the assumption of steady state conditions.

We computed line densities 300 m downstream of the source for the true VCD field and for fields computed with 1D-layer AMFs for different solar zenith and azimuth angles. The VCD cross section are shown in Figure 12. The line densities were multiplied with a wind speed of 9.1 m/s, which is the wind speed at the stack height of 262.5 m in the GRAL simulation.



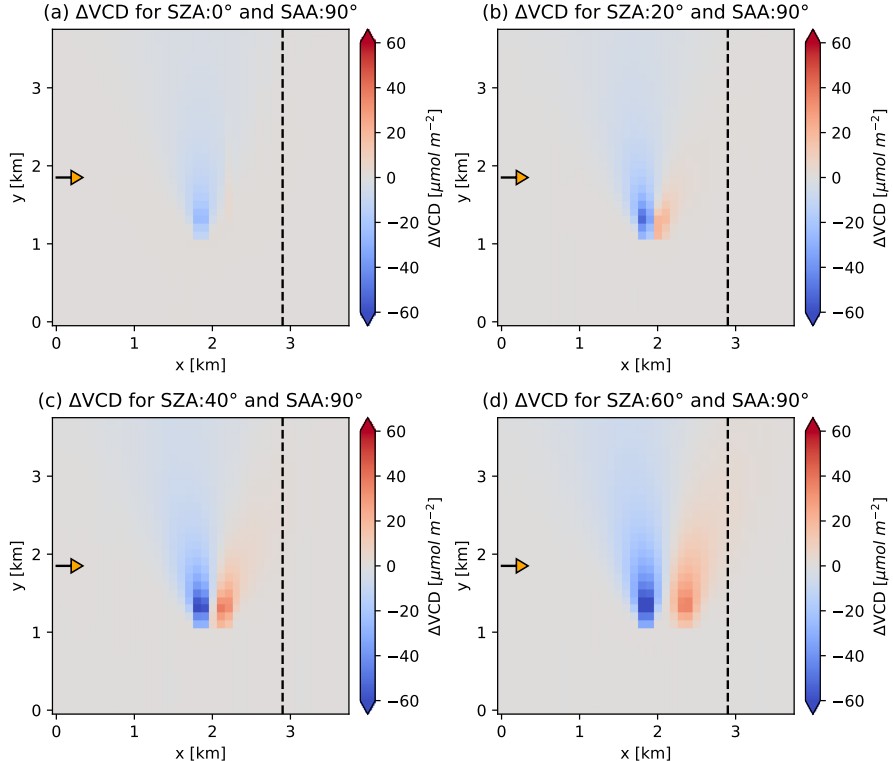

**Figure 10.** Absolute difference between total VCD from synthetic SCD and 1D box AMF with solar zenith angles (SZA) of (a) 0° (b) 20°, (c) 40° and (d) 60° and the true total VCD.

Table 2 summarizes the computed line densities and fluxes for the different scenarios. In all scenarios, emissions were significantly underestimated by 9-37% (relative to the true VCD) depending on the solar azimuth and zenith angle. Note that the emission estimation for the true VCD is slightly higher than the emission input for the dispersion model due to simplification of the mass balance approach, which does not account for the vertical variability in wind speeds across the plume. The bias in the plume emission estimation using 1D-layer AMFs generally increases with solar zenith angle. This bias also depends on the solar azimuth angle. The largest bias occurs, when the relative azimuth angle is 90° or 270°, i.e. the instrument is flying towards and away from the sun.

## 6   Conclusions

This study demonstrates the importance of 3D radiative transfer effects for a range of trace gas remote sensing applications such as ground-based MAX-DOAS and airborne imaging spectroscopy. To study these effects, 1D-layer and 3D-box air mass factors (AMFs) were implemented in the Monte Carlo solver MYSTIC of the radiative transfer model (RTM) libRadtran. The





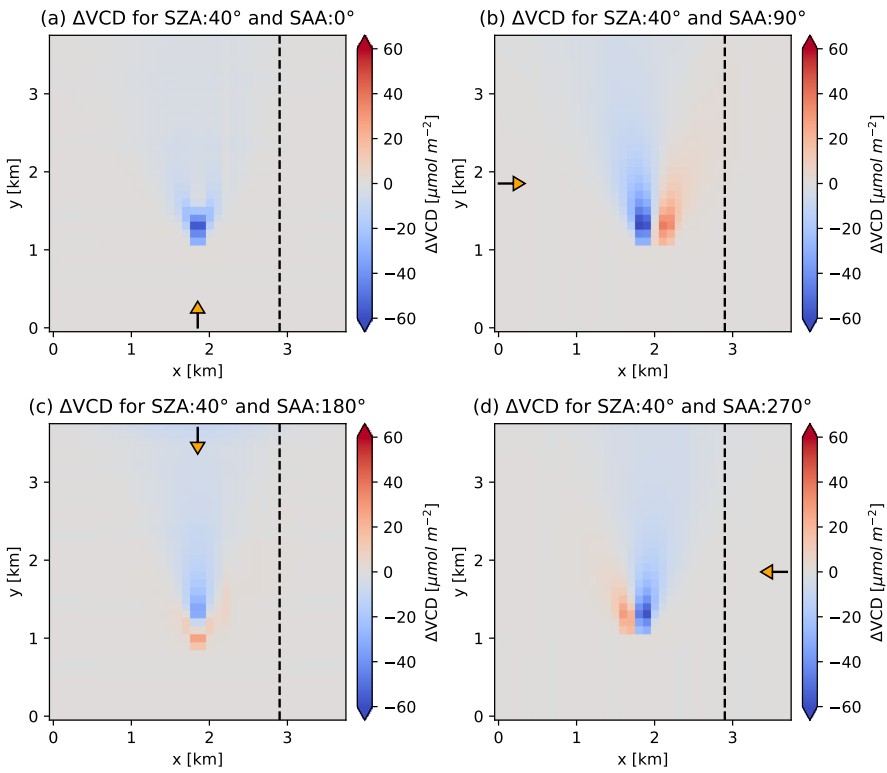

**Figure 11.** Absolute difference between total VCD from synthetic SCD and 1D box AMF with solar azimuth angle of (a) 0°, (b) 90°, (c) 180° and (d) 270° and the true total VCD.

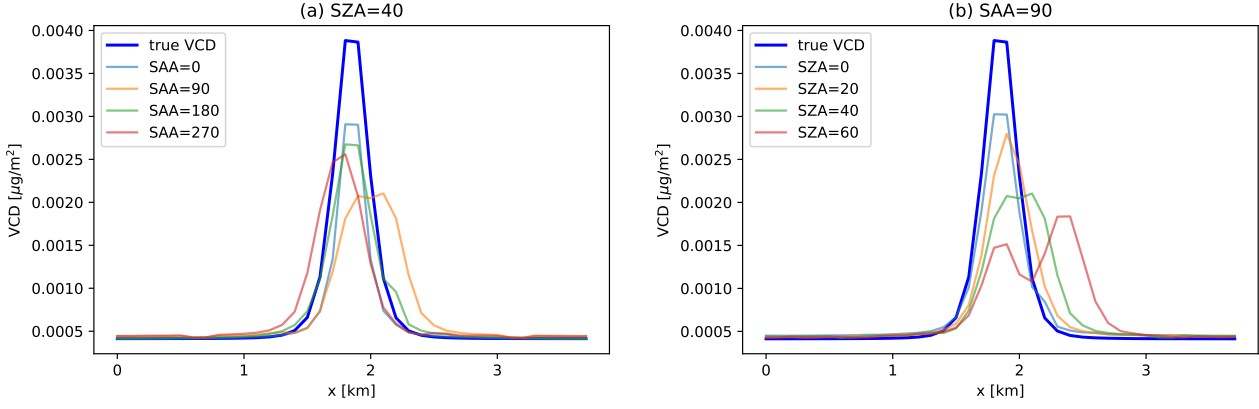

**Figure 12.** Plume VCD cross section at y=1.6 km (0.3 km downstream of the plume) for (a) the sun at SZA=40° with different SAAs and (b) for the sun in the west with different SZAs





computation of AMFs is a central component in most trace gas retrieval algorithms to convert observed slant column densities (SCDs) into vertical column densities (VCDs), but so far these algorithms were limited to 1D RTMs. In case of a horizontally homogeneous atmosphere and in plane parallel geometry, the 3D-box and 1D-layer AMFs perfectly agree within the statistical
noise of the Monte Carlo method. They also agree very well with 1D-layer AMFs calculated with other RTMs presented in a previous model intercomparison study by Wagner et al. (2007).

The importance of 3D effects was demonstrated for two examples. For a ground based MAX-DOAS instrument, we showed that 3D-box AMFs are highest along the line of sight of the instrument (representing photons that have mostly scattered only once), but that the contribution from outside is not negligible and depends on sun position and aerosol optical depth.
The spatial distribution of the vertically integrated 3D-box AMFs depends on the sun position, which can be important for interpreting MAX-DOAS observations, especially in urban areas or, more generally, in the vicinity of pollution sources. The spatial variability of the $NO_2$ distribution in the context of the MAX-DOAS instrument can affect the retrieval differently at different times of the day.

As second example, trace gas retrievals were studied for an airborne imaging spectrometer using simulations of a $NO_2$ plume
emitted by a stack. We showed that when using 1D-layer AMFs, the $NO_2$ VCDs in the plume were significantly underestimated (up to 58%), and that the position of the plume was artificially shifted towards the aircraft. Furthermore, integrals of the $NO_2$ enhancement in across-plume direction (line densities) were also biased, which results in an underestimation of the $NO_2$ emissions from the stack when using a mass-balance approach. Using 1D-layer AMFs induces systematic errors even if the $NO_2$ profile above the ground pixels is known accurately, because a 1D RTM fails to properly represent the complex light path,
which is required if the trace gas field is not horizontally homogeneous.

Our study showed that even for simple examples, 3D effects are not negligible if the trace gas field has a high spatial variability. This finding is particularly relevant for ground-based and airborne remote sensing in cities, where considering 3D effects is likely indispensable to reduce systematic errors. 3D effects are also important for tomographic inversion (e.g. Frins et al., 2006; Kazahaya et al., 2008; Casaballe et al., 2020) where the application 3D-box AMFs will minimise errors caused
by the use of pure geometric assumptions. The high spatial resolution of the next generation of satellite instruments might make it necessary to also consider 3D effects for space-based trace gas remote sensing. Especially when considering imaging spectrometers with very high spatial resolution to estimate emissions (e.g. Strandgren et al., 2020), 3D radiative transfer effects should be considered and studied. However, since 3D radiative transfer calculations are computationally expensive, efficient methods need to be developed for operational applications that provide an appropriate balance between accuracy and
computational cost.

*Code availability.* The libRadtran package including the 1D version of MYSTIC is freely available on www.libradtran.org, the 3D MYSTIC code is available upon request to Claudia Emde (claudia.emde@lmu.de) and the other used codes are available upon request to the corresponding author.





*Data availability.* The data is available upon request to Marc Schwärzel (marc.schwaerzel@empa.ch) for the discussion paper and will be
made available on https://zenodo.org for the final revised paper.

*Author contributions.* MS implemented the 3D box AMF module and validated the implementation, designed and simulated the 3D scenarios
and wrote the manuscript with input from all co-authors. CE implemented 1D-layer AMF module, implemented the 3D box AMFs together
with MS, provided assistance with study design and reviewed the manuscript. TW provided the 1D-layer AMF data used for the validation
and reviewed the manuscript. DB, BB and AB provided critical feedback to the study and reviewed the manuscript, RM conducted and
provided the GRAL simulation, GK supervised the study, designed together with MS the case studies and reviewed the manuscript.

*Competing interests.* The authors declare that they have no conflict of interest.

*Acknowledgements.* This study was conducted as part of the HighNOCS project funded by the Swiss National Science Foundation (SNSF)
under project number 172533.



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
