# Peer review of "Three-dimensional radiative transfer effects on airborne, satellite and ground-based trace gas remote sensing"

_Atmospheric Measurement Techniques, 2020_

## Referee Comment (RC1) · Frederik Tack (Referee) · 12 Jun 2020

**Review of: Three-dimensional radiative transfer effects on airborne, satellite and ground-based trace gas remote sensing (Schwaerzel et al., 2020)**

The manuscript discusses the study of 3D RT/geometric effects on trace gas remote sensing. The 3D distribution of the instrument sensitivity to a certain pollutant is a very relevant problem when observing a heterogeneous atmosphere at high resolution from ground-based, airborne, spaceborne instruments. 3D-BOX AMFs are implemented into the Monte Carlo MYSTIC RTM and validated against 1D-layer AMFs computed by other RTMs. The geometric effects are demonstrated based on a ground-based and airborne case with a focus on $NO_2$ and based on simulated data. The scientific content of the paper fits well within the scope of AMT and will be valuable to a wide community. The manuscript is well-written and generally well-structured. Therefore I highly recommend its publication in AMT. However, some revisions (detailed below) need to be conducted in the paper before publication.

**General comments**

-On p.3, l.43 the authors state that the assumption of horizontal homogeneity (1D-BOX AMF) is not valid in polluted environments. In theory, I fully agree with this statement. However, in case of real-world observations I'm doubting that all relevant data will be available for most regions in order to fully benefit of the 3D-BOX AMF calculations. It requires accurate and high-resolution 3D trace gas and aerosol fields, while it is now often already difficult to get the proper high-resolution a priori for the 1D layer geometry. This has not been discussed in the paper. It would be an added-value to add a paragraph, e.g. in the conclusion, to discuss what is currently missing or needed in order to fully benefit of the 3D BOX-AMF geometry in case of real-world observations.

-I assume it is most likely subject of a future study but I'm also strongly interested to see the impact on total AMFs and VCDs when using 3D BOX-AMFs instead of 'typical" 1D layer AMFs in case of real-world MAX-DOAS and airborne/spaceborne observations. In case this is subject of future work, it could be mentioned in the conclusion.

-A quantification of the horizontal smoothing of the plume due to geometric effects when 1-d layer AMFs are used would be an added-value in addition to Fig. 10 and 11. For example, a table could be added quantifying the horizontal smoothing for different scenarios of SZA, VZA, plume size and plume altitude.

-The reader might also be interested to get some info on absolute computation time and difference between MYSTIC 1D layer AMF and MYSTIC 3D box AMF computation time for a typical satellite or airborne scene.

**Minor comments**

-p.2, l.35 I understand you want to make a distinction between layer-AMFs (1D) and BOX-AMFs (3D). However, in past studies eg Wagner et al. 2007, "BOX-AMFs" were used for what is defined in this work as layer-AMFs. I would add a short statement to clarify. I noticed that you clarify this later on in p.5. I propose to switch it to the introduction.

-p.3, l.68: SCD and VCD acronyms were already defined in the introduction. No need to do it here again. There is some repetition here as well like explaining again what a VCD and AMF is. I suggest to remove it from the introduction or remove it here.

-In Fig. 8b I would expect the two SCD maxima to be east and west of the true VCD at first glance (also when looking at Fig. 9). However, they both seem to be east of the true VCD, with the most western SCD maximum falling together with the VCD maximum. Or do you assume VZA is 0°? In that case it isn't consistent with the example in Fig. 7. Could you please clarify?

**Technical corrections**

-p.1, l.6: MYSTIC acronym stands for …?

-p.2, l.45: I would add "at high resolution" after "trace gas remote sensing"

-p.3, l.63: It depends also on the molecule and aerosol properties (e.g. SSA)

-p.3, l.75: remove "and" after VCD

-p.5, l.87: a**n** aircraft

-p.5, l.88: sensitivity to $NO_2$ → it is a general discussion. I suggest replacing "$NO_2$" by "the trace gas - under investigation"

-p.5, l.110: "computationally efficiency" → computational

-p.6, l.133:  577nm → add space

-p.6, l.137: mostly 1000m resolution → maybe more clear to describe it as a layer thickness instead of vertical resolution.

-p.8, l.161: The upper row of Figures 3 **(scenario at 577 nm)** and 4 **(scenario at 360 nm)**

-p.8, l.162: show**s**

-p.11, l.13: for → at

-p.13, l.257: For clarity, mention explicitly GRAL is a dispersion model, eg. Graz Lagrangian dispersion model

-p.14, l.289: planet boundary layer (PBL) -> boundary layer has occurred many times earlier in the paper. Please explain PBL acronym at first occurrence and use the acronym in the continuation of the work.

-p.15, l.294: SCCs→ SCDs

-p.16, l.302: …3 instrument zenith angles -- > 3 viewing zenith angles (VZA)

-p.16, l.310: (Fig 8 2nd row) → (Fig. 8d, 8e, 8f) (same for line 312)

-p.18, l.339: The VCD cross section**s**

-p.21; l.374: the application **of**

-Figure 2: The spherical regression line and points are not clear at all. Maybe consider having two scatter plots. However, the main message of the plot stays clear based on the 5% deviation lines

-Figure 5 caption: Decay of vertically integrated AMFs with distance to the instrument (c) → please add "is visualized" after "instrument"

-Figure 8: …located at x=19 km and y=13 km. → should be x=1.9 km and y = 1.3 km?

-Figure 12: maybe put "$NO_2$ concentration" instead of "VCD" in plot and caption.

-Table 2: Maybe better to give RAA instead of SAA in order to be consistent with the discussion at the end of p.19

Section 6 (Conclusion): I assume there is no need to define acronyms again here, e.g. AMF, RTM, VCD, SCD, etc.

---

## Referee Comment (RC2) · Anonymous Referee #2 · 14 Jun 2020

**1 Overview**

Schwaerzel et al. use a 3D radiative transfer model to investigate errors induced by the 1D assumptions typically made by atmospheric retrievals for two specific cases - a ground based MAX-DOAS observation and hypothetical plume observation from an airborne pushbroom spectrometer. They first validate the 3D RTM against by simulating horizontally homogeneous conditions. They then investigate the potential 1D induced errors associated with sub-grid variability in the retrieval gas target for the two case studies.

The paper is clearly written and is definitely suitable subject matter for AMT. I highly recommend publication after the following comments are addressed.

**2  General Comments**

**Pg3, Ln43:** The first review pointed out that it would be difficult to obtain the relevant profile information required to use the 3D scattering weight information. Turning this problem on its head, I think it would be worthwhile commenting on the potential to constrain the horizontal gas distribution if you could scan the instrument azimuthally i.e. is there enough information present to invert concentrations radially in the same manner that different viewing zeniths can be used to partially infer the vertical profile. I think this was alluded to in the conclusions but could be expanded on.

**P3, Ln 190:** *"This hypothesis could be tested by including more streams"*
I agree with the reasoning, but wouldn't it be easy to rerun the SCIATRAN simulation with a higher stream number just to confirm that it converges towards the monte carlo method so you can make a more definite statement?

**Pg 13, L239:** The discussion on the line-of-sight sensitivity in this paragraph is useful for gaining some physical intuition for the observations. It would be useful to more systematically explore this as a function of AOD/view geometry, to provide guidelines for situations that permit the interpretation of when the majority of photons are coming to the instrument by single scatter into the path of the detector. It is possibly more relevant to only consider the line-of-sight within the assumed boundary layer, where the largest horizontal variation of $NO_2$ is expected to be.
**Pg 13, L245** It would be nice to see a concrete example of assuming the horizontal homogeneity from the computed box-AMFs by using high resolution CTM output (if the authors have access to any). Although the resolution still may be slightly too coarse, NO2 data from the NASA GEOS-5 nature run is publicly available and may be possibly worth trying (https://portal.nccs.nasa.gov/datashare/G5NR-Chem/Heracles/12.5km/DATA/0.125_deg)
I realize it may not be computationally feasible to run multiple scenarios with the correct solar geometries for particular scenes, but even applying the computed box-AMFs in Figure 8 to ground locations across a set of CTM fields may shed some light on what diurnal/synoptic-time scales

**3 Minor Comments/Corrections**

I think somewhere it would be helpful to mention how long the monte carlo calculations take perhaps in a until of time/photons to get an idea of how long the scene calculations take

**Pg 3, Ln 69:** The following equation for SCD more accurately captures the way you have described it

$$SCD = \frac{1}{n} \sum_{i=1}^{n} \int_{path_i} c(l)dl \qquad (1)$$

[Figure]

**Pg 5, Ln 98:** I think MCARaTs is another Monte Carlo RTM with the capability of computing AMFs (https://sites.google.com/site/mcarats/home)

**Pg 15, Ln 293** *"The SCCs are larger than the VCDs (panel a) because the AMFs (panel c) are generally larger than 1"* SCC -> SCD. Also, is this a tautology?

**Pg 17, Fig. 8** x=19 km and y=13 km -> x=1.9 km and y=1.3 km

---

## Author Comment (AC1) · 7 Jul 2020

**Three-dimensional radiative transfer effects on airborne, satellite and ground-based trace gas remote sensing**

Marc Schwaerzel et al.

**Response to the Reviewer's Comments**

We thank the two reviewers for their positive comments, critical assessment and useful points to improve the quality of our paper. In the following we address their concerns point by point. Changes in the paper are shown in blue. We hope we clarified all concerns and that the revised manuscript has improved.
* * *
**5 Reviewer 1**

**Reviewer Point P 1.1** — On p.3, l.43 the authors state that the assumption of horizontal homogeneity (1D-BOX AMF) is not valid in polluted environments. In theory, I fully agree with this statement. However, in case of real-world observations I'm doubting that all relevant data will be available for most regions in order to fully benefit of the 3D-BOX AMF calculations. It requires accurate and high-resolution 3D trace gas and aerosol fields, while it is now often already difficult to get the proper high-resolution a priori for the 1D layer geometry. This has not been discussed in the paper. It would be an added-value to add a paragraph, e.g. in the conclusion, to discuss what is currently missing or needed in order to fully benefit of the 3D BOX-AMF geometry in case of real-world observations.

**Reply**: We agree with the reviewer that taking full advantage of 3D-box AMFs requires reliable 3D information that is generally difficult to obtain. We will address this question in further studies, where we plan to use city-scale dispersion models to obtain the 3D fields. We added a paragraph in the conclusion on the particular need of 3D box-air mass factors calculations over a polluted area.

> To fully benefit of 3D-box AMFs, 3D radiative transfer calculations require high-resolution 3D distributions of trace gases and aerosols to calculate the total AMF. Such fields are generally difficult to obtain. In a follow-up study we plan to use 3D $NO_2$ fields from a building-resolving urban air quality model (Berchet et al., 2017) with a detailed representation of both near-surface and elevated (stack) emission sources to further analyze the added value of 3D-box AMFs.

**Reviewer Point P 1.2** — I assume it is most likely subject of a future study but I'm also strongly interested to see the impact on total AMFs and VCDs when using 3D BOX-AMFs instead of 'typical" 1D layer AMFs in case of real-world MAX-DOAS and airborne/spaceborne observations. In case this is subject of future work, it could be mentioned in the conclusion.

**Reply**: We also think impact on total AMFs and VCDs an important point. The effects of using 3D-box AMFs on total AMFs and on VCDs and the horizontal smoothing observed with 3D-box AMFs will be addressed in further studies as described in Point P.1.1.

**Reviewer Point P 1.3** — A quantification of the horizontal smoothing of the plume due to geometric effects when 1-d layer AMFs are used would be an added-value in addition to Fig. 10 and 11. For example, a table could be added quantifying the horizontal smoothing for different scenarios of SZA, VZA, plume size and plume altitude.

**Reply**: Since horizontal smoothing is an important, but also quite complex topic, we plan to address it in a follow-up study in much more detail than possible here. We added a sentence to the conclusion.

> This finding is particularly relevant for ground-based and airborne remote sensing in cities, where considering 3D effects is likely indispensable to reduce systematic errors. This will be addressed in a follow-up study where also the potential impact of 3D radiative transfer effects on the horizontal smoothing of the retrieved trace gas fields will be studied.

**Reviewer Point P 1.4** — The reader might also be interested to get some info on absolute computation time and difference between MYSTIC 1D layer AMF and MYSTIC 3D box AMF computation time for a typical satellite or airborne scene.

**Reply**: We considered that point and added the following paragraph to section 5.1

> The computational cost of calculating 3D-box AMFs is considerably larger than for 1D-layer AMFs. The computational time for calculating 3D-box AMFs for the scenarios here (see Table 1 with SZA=20°, SAA=90°, VAA=90° and VZA=2°) is around 218 seconds with 1 million photons using a single core of our local machine (Intel Xeon W-2175 CPU @ 2.5 GHz). The computational time for the corresponding 1D-layer AMFs is only about 4 seconds with 1 million photons. Note, however, that even less photons would be sufficient to obtain a similar noise level as for the 3D-box AMFs.

**Reviewer Point P 1.5** — p.2, l.35 I understand you want to make a distinction between layer-AMFs (1D) and BOX-AMFs (3D). However, in past studies eg Wagner et al. 2007, "BOX-AMFs" were used for what is defined in

this work as layer-AMFs. I would add a short statement to clarify. I noticed that you clarify this later on in p.5. I propose to switch it to the introduction.

**Reply**: We moved the clarifying comment to the introduction.

55    Notice that in previous studies (e.g. Rozanov and Rozanov, 2010) 1D-layer AMF were some times referred to as box AMFs. In this study, we will use the terms 1D-layer and 3D-box AMFs to clearly distinguish between them.

**Reviewer Point P 1.6** — p.3, l.68: SCD and VCD acronyms were already defined in the introduction. No need to do it here again. There is some repetition here as well like explaining again what a VCD and AMF is. I suggest

60    to remove it from the introduction or remove it here.

**Reply**: We deleted the repetitions and replaced the acronym definitions by the acronyms in the method section.

**Reviewer Point P 1.7** — In Fig. 8b I would expect the two SCD maxima to be east and west of the true VCD at first glance (also when looking at Fig. 9). However, they both seem to be east of the true VCD, with the most

65    western SCD maximum falling together with the VCD maximum. Or do you assume VZA is 0°? In that case it isn't consistent with the example in Fig. 7. Could you please clarify?

**Reply**: The example shown in Figure 7 uses a different observation geometry than the one used in Figure 8. We updated Figure 7 to show an example from Figure 8 now. We also modified Figure 9 to match the explanation.

70    **Reviewer Point P 1.8** — p.1, l.6: MYSTIC acronym stands for ...?

**Reply**: MYSTIC stands for Monte carlo code for the phYSically correct Tracing of photons In Cloudy atmospheres. We added this in the abstract.

...,  we implemented 1D-layer and 3D-box AMFs into the Monte carlo code for the phYSically correct Tracing of photons In Cloudy atmospheres (MYSTIC), a solver of the libRadtran radiative transfer model

75    (RTM).

**Reviewer Point P 1.9** — p.2, l.45: I would add "at high resolution" after "trace gas remote sensing"

**Reply**: We modified the text accordingly.

**Reviewer Point P 1.10** — p.3, l.63: It depends also on the molecule and aerosol properties (e.g. SSA)

**Reply**: We modified the sentence as follows:

80    Atmospheric scattering and absorption is determined by the distribution and properties of molecules, aerosols and clouds, and depends on the wavelength of the radiation. Molecular scattering is particularly important in the UV range of the spectrum.

**Reviewer Point P 1.11** — p.3, l.75: remove "and" after VCD

**Reply**: We corrected the typo.

85  **Reviewer Point P 1.12** — p.5, l.87: an aircraft

**Reply**: We corrected the typo.

**Reviewer Point P 1.13** — p.5, l.88: sensitivity to NO2 -> it is a general discussion. I suggest replacing "NO2" by "the trace gas under investigation"

**Reply**: We changed the text as suggested.

90    In this case, the AMF can be interpreted as the instrument sensitivity to the trace gas under investigation for measuring that specific VCD.

**Reviewer Point P 1.14** — p.5, l.110: "computationally efficiency" -> computational

**Reply**: We corrected the mistake.

**Reviewer Point P 1.15** — p.6, l.133: 577nm -> add space

95  **Reply**: We corrected the typo.

**Reviewer Point P 1.16** — p.6, l.137: mostly 1000m resolution -> maybe more clear to describe it as a layer thickness instead of vertical resolution.

**Reply**: We changed the text as suggested.

For the simulations 17 vertical layers were used with a thickness of $100\,\mathrm{m}$ below $1000\,\mathrm{m}$ and a thickness
100    of mostly $1000\,\mathrm{m}$ above (see Table 1 in Wagner et al., 2007).

**Reviewer Point P 1.17** — p.8, l.161: The upper row of Figures 3 (scenario at 577 nm) and 4 (scenario at 360 nm)

**Reply**: We changed the text as suggested.

The upper row of Figures 3 (scenario at 577 nm) and 4 (scenario at 360 nm) shows MYSTIC 1D-layer AMF profiles for the selected scenarios with a low elevation angle of 3° and a high elevation angle of 90° (zenith) without and with aerosols, respectively.

**Reviewer Point P 1.18** — p.8, l.162: shows

**Reply**: We corrected the typo.

**Reviewer Point P 1.19** — p.11, l.13: for -> at

**Reply**: We implemented the suggested change.

**Reviewer Point P 1.20** — p.13, l.257: For clarity, mention explicitly GRAL is a dispersion model, eg. Graz Lagrangian dispersion model

**Reply**: We changed the text as suggested.

The $NO_2$ plume was computed with the Graz Lagrangian dispersion model (GRAL) (Oettl, 2015) for a 262.5 m tall stack located at x=1.9 km and y=1.3 km.

**Reviewer Point P 1.21** — p.14, l.289: planet boundary layer (PBL) -> boundary layer has occurred many times earlier in the paper. Please explain PBL acronym at first occurrence and use the acronym in the continuation of the work.

**Reply**: We changed the manuscript accordingly.

**Reviewer Point P 1.22** — p.15, l.294: SCCs->SCDs

**Reply**: We corrected the typo.

**Reviewer Point P 1.23** — p.16, l.302: . . . 3 instrument zenith angles – > 3 viewing zenith angles (VZA)

**Reply**: We changed the text accordingly.

This is further illustrated in Fig. 9, where 2 of the 3 illustrated direct paths (i.e. 3 viewing zenith angles) cross the $NO_2$ maximum (main photon path (1) and (3) in Fig. 9)

**Reviewer Point P 1.24** — p.16, l.310: (Fig 8 2nd row) ->(Fig. 8d, 8e, 8f) (same for line 312)

**Reply**: We changed the text accordingly.

**Reviewer Point P 1.25** — p.18, l.339: The VCD cross sections

**Reply**: We corrected the typo.

130 **Reviewer Point P 1.26** — p.21; l.374: the application of

**Reply**: We corrected the typo.

**Reviewer Point P 1.27** — The spherical regression line and points are not clear at all. Maybe consider having two scatter plots. However, the main message of the plot stays clear based on the 5% deviation lines

**Reply**: We separated the one scatter plot in two scatter plots.

135

**Reviewer Point P 1.28** — Figure 5 caption: Decay of vertically integrated AMFs with distance to the instrument (c) -> pease add"is visualized" after "instrument"

**Reply**: We modified the caption as suggested.

**Reviewer Point P 1.29** — Figure 8: ...located at x=19 km and y=13 km. -> should be x=1.9 km and y = 1.3 140 km?

**Reply**: We corrected the typo.

**Reviewer Point P 1.30** — Figure 12: maybe put "NO2 concentration"instead of "VCD" in plot and caption.

**Reply**: We modified the plot axis label to $NO_2$ column densities $[\mu mol/m^2]$ and kept VCD in the caption because $NO_2$ concentration implies a point quantity. To be consistent with other figures, we changed the unit to $\mu mol/m^2$.

145 **Reviewer Point P 1.31** — Table 2: Maybe better to give RAA instead of SAA in order to be consistent with the discussion at the end of p.19

**Reply**: RAA would not work in the table, because for pixels on the east of the instrument, have another viewing azimuth angle than pixels on the west for the same scene. We modified the text to be correct and consistent with the table.

**Reviewer Point P 1.32** — Section 6 (Conclusion): I assume there is no need to define acronyms again here, e.g. 150 AMF, RTM, VCD, SCD, etc.

**Reply**: We modified the text as suggested.

**References**

Berchet, A., Zink, K., Muller, C., Oettl, D., Brunner, J., Emmenegger, L., and Brunner, D.: A cost-effective method for simulating city-wide air flow and pollutant dispersion at building resolving scale, Atmospheric environment, 158, 181–196, 2017.

Oettl, D.: Quality assurance of the prognostic, microscale wind-field model GRAL 14.8 using wind-tunnel data provided by the German VDI guideline 3783-9, Journal of Wind Engineering and Industrial Aerodynamics, 142, 104–110, https://doi.org/10.1016/j.jweia.2015.03.014, http://www.sciencedirect.com/science/article/pii/S0167610515000744, 2015.

Rozanov, V. and Rozanov, A.: Differential optical absorption spectroscopy (DOAS) and air mass factor concept for a multiply scattering vertically inhomogeneous medium: theoretical consideration, Atmospheric Measurement Techniques, 3, 751–780, 2010.

Wagner, T., Burrows, J. P., Deutschmann, T., Dix, B., von Friedeburg, C., Frieß, U., Hendrick, F., Heue, K.-P., Irie, H., Iwabuchi, H., Kanaya, Y., Keller, J., McLinden, C. A., Oetjen, H., Palazzi, E., Petritoli, A., Platt, U., Postylyakov, O., Pukite, J., Richter, A., van Roozendael, M., Rozanov, A., Rozanov, V., Sinreich, R., Sanghavi, S., and Wittrock, F.: Comparison of box-air-mass-factors and radiances for Multiple-Axis Differential Optical Absorption Spectroscopy (MAX-DOAS) geometries calculated from different UV/visible radiative transfer models, Atmospheric Chemistry and Physics, 7, 1809–1833, 2007.

---

## Author Comment (AC2) · 7 Jul 2020

**Three-dimensional radiative transfer effects on airborne, satellite and ground-based trace gas remote sensing**

Marc Schwaerzel et al.

**Response to the Reviewer's Comments**

We thank the two reviewers for their positive comments, critical assessment and useful points to improve the quality of our paper. In the following we address their concerns point by point. Changes in the paper are shown in blue. We hope we clarified all concerns and that the revised manuscript has improved.
* * *
5 ## Reviewer 2

**Reviewer Point P 2.1** — Pg3, Ln43: The first review pointed out that it would be difficult to obtain the relevant profile information required to use the 3D scattering weight information. Turning this problem on its head, I think it would be worthwhile commenting on the potential to constrain the horizontal gas distribution if you could scan the instrument azimuthally i.e. is there enough information present to invert concentrations radially in the same
10 manner that different viewing zeniths can be used to partially infer the vertical profile. I think this was alluded to in the conclusions but could be expanded on.

**Reply**: This is an important point. We added a sentence on this approach in the conclusions:

> On the other hand, measuring different azimuth angles with a MAX-DOAS instrument could be used to constrain the 3D fields of trace gases (e.g. Dimitropoulou et al., 2019).

15 **Reviewer Point P 2.2** — P3, Ln 190:"This hypothesis could be tested by including more streams "I agree with the reasoning, but wouldn't it be easy to rerun the SCIATRAN simulation with a higher stream number just to confirm that it converges towards the monte carlo method so you can make a more definite statement?

**Reply**: Since we used 1D-layer AMFs from a previous study (Wagner et al., 2007), we have neither the SCIATRAN executable used in that study nor the exact input files to replicate the simulations. We therefore think that the effort
20 required to set up SCIATRAN just for this one test is too large, also because the difference only affects altitudes that are less relevant for our study.

**Reviewer Point P 2.3** — Pg 13, L239:The discussion on the line-of-sight sensitivity in this paragraph is useful for gaining some physical intuition for the observations. It would be useful to more systematically explore this as a function of AOD/view geometry, to provide guidelines for situations that permit the interpretation of when the majority of photons are coming to the instrument by single scatter into the path of the detector. It is possibly more relevant to only consider the line-of-sight within the assumed boundary layer, where the largest horizontal variation of NO2 is expected to be.

**Reply**: This is a valid point and we agree that such information could be useful for further studies. In this paper, we want to show the general importance of 3D in radiative transfer modelling for trace gas retrieval. We feel that quantifying effects of particular conditions (i.e. input parameters) and particular measurement setting on an instrument sensitivity would be complex and out of scope of this study.

In section 4 (Figure 6) we already considered the line-of-sight only within the boundary layer.

**Reviewer Point P 2.4** — I think somewhere it would be helpful to mention how long the monte carlo calculation stake perhaps in a until of time/photons to get an idea of how long the scene calculation stake

**Reply**: We addressed this question in the response to the first reviewer. We added a paragraph in section 5.1.

The computational cost of calculating 3D-box AMFs is considerably larger than for 1D-layer AMFs. The computational time for calculating 3D-box AMFs for the scenarios here (see Table 1 with SZA=20°, SAA=90°, VAA=90° and VZA=2°) is around 218 seconds with 1 million photons using a single core of our local machine (Intel Xeon W-2175 CPU @ 2.5 GHz). The computational time for the corresponding 1D-layer AMFs is only about 4 seconds with 1 million photons. Note, however, that even less photons would be sufficient to obtain a similar noise level as for the 3D-box AMFs.

**Reviewer Point P 2.5** — Pg 3, Ln 69: The following equation for SCD more accurately captures the way you have described it

$$SCD = \frac{1}{n} \sum_{i=1}^{n} \int_{path_i} c(l)dl \tag{1}$$

**Reply**: Our methods section starts from general equations to specific equations. The suggested equation already implies a solution calculated with a Monte-Carlo solvers. Therefore we did not apply the suggested change.

**Reviewer Point P 2.6** — Pg 5, Ln 98:I think MCARaTs is another Monte Carlo RTM with the capability ofcomputing AMFs (https://sites.google.com/site/mcarats/home)

**Reply**: The MCARaTs RTM computes 1D-layer air mass factors and was part of the Wagner et al. (2007) RTM inter-comparison study. However, we didn't find any indication that it is also able of computing/outputting 3D-box AMFs. We added two citations of the model in the introduction.

> In the past decades, numerous RTMs have been developed with the possibility to calculate one-dimensional layer AMFs (e.g. Berk et al., 1999; Postylyakov, 2004; Rozanov et al., 2005; Wagner et al., 2007; Spurr et al., 2001; Iwabuchi, 2006; Iwabuchi and Okamura, 2017). The computation of layer AMFs is implemented in most trace gas retrieval algorithms for satellite and ground-based observations applied today (Boersma et al., 2011; Irie et al., 2011; Wenig et al., 2008; Wu et al., 2013).

**Reviewer Point P 2.7** — Pg 15, Ln 293: The SCCs are larger than the VCDs (panel a) because the AMFs(panel c) are generally larger than 1"SCC -> SCD. Also, is this a tautology?

**Reply**: We corrected the typo and do not think the sentence is a tautology.

**Reviewer Point P 2.8** — Pg 17, Fig. 8x=19 km and y=13 km -> x=1.9 km and y=1.3 km

**Reply**: We corrected the typo.

**References**

Berk, A., Anderson, G. P., Bernstein, L. S., Acharya, P. K., Dothe, H., Matthew, M. W., Adler-Golden, S. M., Chetwynd Jr, J. H., Richtsmeier, S. C., Pukall, B., et al.: MODTRAN4 radiative transfer modeling for atmospheric correction, in: Optical spectroscopic techniques and instrumentation for atmospheric and space research III, vol. 3756, pp. 348–353, International Society for Optics and Photonics, 1999.

Boersma, K. F., Eskes, H. J., Dirksen, R. J., van der A, R. J., Veefkind, J. P., Stammes, P., Huijnen, V., Kleipool, Q. L., Sneep, M., Claas, J., Leitão, J., Richter, A., Zhou, Y., and Brunner, D.: An improved tropospheric NO2 column retrieval algorithm for the Ozone Monitoring Instrument, Atmospheric Measurement Techniques, 4, 1905–1928, 2011.

Dimitropoulou, E., Van Roozendael, M., Hendrick, F., Merlaud, A., Tack, F., Fayt, C., Hermans, C., and Pinardi, G.: One year of 3-D MAX-DOAS tropospheric NO2 measurements over Brussels, in: EGU General Assembly Conference Abstracts, EGU General Assembly Conference Abstracts, p. 5874, 2019.

Irie, H., Takashima, H., Kanaya, Y., Boersma, K. F., Gast, L., Wittrock, F., Brunner, D., Zhou, Y., and Van Roozendael, M.: Eight-component retrievals from ground-based MAX-DOAS observations, Atmospheric Measurement Techniques, 4, 1027–1044, http://www.atmos-meas-tech.net/4/1027/2011/, 2011.

Iwabuchi, H.: Efficient Monte Carlo methods for radiative transfer modeling, Journal of the atmospheric sciences, 63, 2324–2339, 2006.

Iwabuchi, H. and Okamura, R.: Multispectral Monte Carlo radiative transfer simulation by the maximum cross-section method, Journal of Quantitative Spectroscopy and Radiative Transfer, 193, 40–46, 2017.

Postylyakov, O.: Radiative transfer model MCC++ with evaluation of weighting functions in spherical atmosphere for use in retrieval algorithms, Advances in Space Research, 34, 721–726, 2004.

Rozanov, A., Rozanov, V., Buchwitz, M., Kokhanovsky, A., and Burrows, J.: SCIATRAN 2.0–A new radiative transfer model for geophysical applications in the 175–2400nm spectral region, Advances in Space Research, 36, 1015–1019, 2005.

Spurr, R., Kurosu, T., and Chance, K.: A linearized discrete ordinate radiative transfer model for atmospheric remote-sensing retrieval, Journal of Quantitative Spectroscopy and Radiative Transfer, 68, 689–735, 2001.

Wagner, T., Burrows, J. P., Deutschmann, T., Dix, B., von Friedeburg, C., Frieß, U., Hendrick, F., Heue, K.-P., Irie, H., Iwabuchi, H., Kanaya, Y., Keller, J., McLinden, C. A., Oetjen, H., Palazzi, E., Petritoli, A., Platt, U., Postylyakov, O., Pukite, J., Richter, A., van Roozendael, M., Rozanov, A., Rozanov, V., Sinreich, R., Sanghavi, S., and Wittrock, F.: Comparison of box-air-mass-factors and radiances for Multiple-Axis Differential Optical Absorption Spectroscopy (MAX-DOAS) geometries calculated from different UV/visible radiative transfer models, Atmospheric Chemistry and Physics, 7, 1809–1833, 2007.

Wenig, M. O., Cede, A. M., Bucsela, E. J., Celarier, E. A., Boersma, K. F., Veefkind, J. P., Brinksma, E. J., Gleason, J. F., and Herman, J. R.: Validation of OMI tropospheric NO2 column densities using direct-Sun mode Brewer measurements at NASA Goddard Space Flight Center, Journal of Geophysical Research, 113, D16S45, 2008.

Wu, F., Xie, P., Li, A., Chan, K., Hartl, A., Wang, Y., Si, F., Zeng, Y., Qin, M., Xu, J., et al.: Observations of SO2 and NO2 by mobile DOAS in the Guangzhou eastern area during the Asian Games 2010, Atmospheric Measurement Techniques, 6, 2277–2292, 2013.

---

## Author Response (AR2)

Dear Folkert Boersma,

Thank you for accepting our manuscript for publication in AMT.

We proceeded to the technical corrections as following:

1. Remove the word 'satellite' from the title. The paper is about 3-D effects on airborne and ground-based trace gas retrievals, and not about satellite remote sensing.
I removed the word 'satellite' from the title.

2. The caption of Figure 1 needs to make clear what text refers to panel (a), (b), etc.
I modified the caption of Figure 1 according to the suggestion.

3. In the captions of Figures 2 and 3, please make clear that the AMFs shown hold for ground-based viewing geometries.
I modified the caption of Figure 2, 3 and 4 according to your suggestion by making clear, the AMFs were simulated for MAX-DOAS observations.

Yours sincerely,
Marc Schwaerzel

[revised manuscript text omitted]